# Effect of Oat Hull as a Source of Insoluble Dietary Fibre on Changes in the Microbial Status of Gastrointestinal Tract in Broiler Chickens

**DOI:** 10.3390/ani12192721

**Published:** 2022-10-10

**Authors:** Patrycja Wróblewska, Tomasz Hikawczuk, Kamil Sierżant, Andrzej Wiliczkiewicz, Anna Szuba-Trznadel

**Affiliations:** 1Department of Animal Nutrition and Feed Science, Wroclaw University of Environmental and Life Sciences, 38 c Chełmońskiego Street, 51-630 Wrocław, Poland; 2Statistical Analysis Centre, Wroclaw Medical University, 2-6 Marcinkowskiego Street, 50-368 Wrocław, Poland

**Keywords:** dietary fibre, oat hull, broiler chickens, microbial status, crop

## Abstract

**Simple Summary:**

The aim of this study was to determine the effect of the addition of oat hull (0–3%) and high amounts of cereal grains to the diets of broiler chickens in terms of the development of the upper gastrointestinal tract, and individual microbial counts in the crop and ileum, and the effect of dietary fibre fractions on microbial changes from the beak to the ileum. The addition of 3% oat hull increased the weights of the proventriculus and gizzard, thereby improving the gizzard barrier function. The presence of dietary fibre and hemicelluloses in diets increased the total aerobic microbial count and *Lactobacillus* spp. bulk in the crop. The presence of insoluble dietary fibre in the diet decreased the total aerobic microbial count and increased the *E. coli* count in the crop. In the ileum, insoluble dietary fibre decreased the *E. coli* count and soluble dietary fibre decreased the total combined yeast and mould count.

**Abstract:**

This study aimed to determine the effect of the addition of oat hull (0–3%) and high amounts of cereal grains to the diet of broiler chickens in terms of the development of the upper gastrointestinal tract, individual microbial counts in the crop and ileum, and the effect of dietary fibre fractions on microbial changes from the beak to the ileum. In the 28 d trial, 162 one-day-old Hubbard Flex male chickens with an average body weight of 44.5 g were randomly allocated to 27 metabolic cages. The experiment consisted of a randomised, one factorial ANCOVA design composed of a covariate with two ANOVA factorial designs containing nine treatments (3 × 3): three sources of cereal grains (maize, wheat, and barley, with a minimum amount of 500 g · kg^−1^, each with nine replications) and three levels of oat hull (0, 1, and 3%, each with nine replications). At the end of the study, 81 chickens (9 in each treatment) were slaughtered to determine the weight of the individual organs and characterise the intestinal microbiota. The application of 3% oat hull to the cereal diets increased the weight of the proventriculus and the gizzard (6.3 and 27.3 g, respectively) in comparison to diets without the addition of this structural component (6.0 and 23.7 g). Higher crop counts of total aerobic microbes (6.29 log CFU · g^−1^) and *Lactobacillus* spp. (4.05 log CFU · g^−1^) were observed in diets containing wheat grain compared with maize (4.62 and 3.55 log CFU · g^−1^, respectively). The main reason for the microorganism’s growth (*p* < 0.05) was the amount of soluble dietary fibre and hemicelluloses present in the diet: total aerobic microbial count (respectively r = 0.918 and r = 0.816) and *Lactobacillus* spp. (respectively r = 0.416 and r = 0.442). Barley diets decreased (*p* < 0.05) *E. coli* counts in the ileum (2.69 log CFU · g^−1^) vs. maize and wheat diets (3.41 and 3.45 log CFU · g^−1^, respectively), mainly due to the increase in the amount of insoluble dietary fibre in the diet (r = −0.462). Reduced total yeast and mould counts in the ileum were also observed (*p* < 0.05) in connection with the presence of soluble dietary fibre and hemicelluloses in diets (r = −0.397 and −0.398, respectively).

## 1. Introduction

Dietary fibre is an important element of the diet and is present not only in the diet of ruminants but also in that of monogastric animals. This nutrient is not degradable by host exogenous enzymes but can be decomposed by microbes [1]. It is treated similarly to a diluent in a diet, and its inclusion in small amounts can improve the growth performance, health, and welfare of birds [2,3]. Benefits are observed when the amount of added structural components rich in insoluble fibre does not exceed 7% of the total with other feed ingredients [4]. Dietary fibre plays two important roles. Firstly, it regulates digestive motility by improving the functioning of the upper gastrointestinal tract (GIT), mainly due to suppressing the multiplication of pathogenic microorganisms and fostering the development of positive bacteria during its retention time in the crop, promoting higher production of HCl in the proventriculus and prolonging the retention time of digesta in the gizzard [5,6,7,8]. Secondly, it regulates microbial status in the lower gastrointestinal tract, especially in the jejunum and ileum [9].

Some contradictions are found regarding the definition of dietary fibre because it can be described more or less precisely depending on its fractions. One way of classifying dietary fibre is by means of a chemical definition, whereby carbohydrates and lignin can be divided into water-soluble and water-insoluble in the case of nonstarch polysaccharides (NSP) [10,11,12]. On the other hand, there is an analytical definition, according to which dietary fibre can be determined using the Van Soest and Wine [13] method, which is commonly used in the case of ruminant fibre analysis, or the Asp method [14], which is used to determine the total dietary fibre and its soluble and insoluble fractions in food or feed [15,16,17]. The procedure for determining dietary fibre by this method results in more accurate analytical results, as it takes into account soluble dietary fibre [18]. This provides information on the contents of total dietary fibre (TDF) as well as its constituent soluble dietary fibre (SDF) and insoluble dietary fibre (IDF). The soluble fraction includes pectin, gums, β-glucan, and hemicelluloses. In contrast, IDF is mainly cellulose [19]. SDF is quickly fermented in the large intestine, which results in increased concentrations of short-chain fatty acids (SCFA). Moreover, changes in the microflora composition are observed when sources of soluble fibre are added to the diet [20]. A small amount of soluble dietary fibre is beneficial for chicken microflora, but excessive proportions lead to a decreased absorption of nutrients and uncontrolled changes in the microflora. Therefore, the dilution effect is used to counteract such changes, which involves the inclusion of components rich in insoluble dietary fibre [3,21].

Oat hull (OH) is a source of insoluble fibre that can improve daily body gain in broiler chickens when high amounts of wheat or barley rich in non-starch polysaccharides are used in feed, especially in young chickens [22,23,24]. In terms of broiler chicken nutrition, oat hull and fibre solubility affect the microflora status in the crop and small intestine due to the retention time and the content of cellulose in the crop and gizzard, which can cause decreases in the crop and gizzard pH, thus creating a barrier for pathogenic bacteria such as *Salmonella* sp., *C. perfringens*, and enterotoxigenic strains of the *E. coli* or *Eimeria* species [9,25,26]. Additionally, insoluble dietary fibre removes pathogenic bacteria from the small intestine mucosal layer, thereby decreasing their chance of adhering to receptors [3]. For many years, there has been an opinion that soluble fibre leads to problems with pathogenic bacteria in the intestines but mainly in the case of wheat or barley grain. Other sources of dietary fibre, such as sugar beet pulp, soybean hull, or rice hull, also help to keep the balance between pathogenic and symbiotic microflora in the intestines; in this case, the soluble fibre (in small amounts) can participate in restoring the mucosal barrier in intestines. [27,28,29]. Dietary fibre from feed also exerts positive effects by increasing the count of the beneficial bacteria *Lactobacillus* sp., which is present in large amounts in the crop, proventriculus, gizzard, and ileum [30].

The purpose of this experiment was to determine the influence of adding different levels of oat hull (0–3%) and cereal grains on the microbial status in the crop and the initial part of the ileum in broiler chickens, the development of the upper gastrointestinal tract, the correlation between the counts of major groups of bacteria in the diet vs. counts in the crop and ileum, and the relation between different fibre fractions and microorganism counts in the crop and ileum.

## 2. Materials and Methods

### 2.1. Birds, Diets, and Nutrition

The experimental protocol was approved by the Local Ethical Review Committee for Animal Experiments in Wrocław, Poland (protocol no. 084/2010). The study was conducted with 162 birds, which were kept in 27 metabolic cages, and was planned as a randomised one-factorial analysis of covariance (ANCOVA) design composed of a covariate with a two-ANOVA factorial design containing nine treatments (3 × 3): three sources of cereal grains (maize, wheat, and barley, each with nine replications) and three levels of oat hull (0, 1, and 3%, each with nine replications, arranged according to the case of each factor as shown in Table 1). The composition of structural carbohydrates in the components is presented in Table 2.

One-day-old male Hubbard Flex chickens with a mean body weight value of 44.5 ± 1.6 g were randomly allocated to 27 metabolic cages. The birds were fed the starter diet (Table 3) until the 28th day of their life.

In all treatments, cereal grains were used at a minimal amount of 50%, and oat hull was added in amounts from 0 to 3% (Table 3). To prepare the diets according to requirements, the European Tables of Energy Values for Poultry were used to calculate the metabolic energy of the feed components [31]. The Polish Requirements of Poultry Nutrition were used to determine the required level of metabolic energy, crude protein, and crude fibre [32]. The energy value of starter diets is stated per 1 kg of feed in the range from 12.16 to 12.25 MJ and crude protein in an amount of 211.2 to 230.7 g. All diets were isoenergetic and isoproteic (Table 4).

Chickens were kept under standard breeding conditions. The temperature on the first day of the experiment was 32 °C, and it was systematically decreased to 20 °C on the 28th day of the birds’ lives. The lighting programme consisted of 20 h of light and 4 h of darkness during the first 10 days. Then, the period of darkness was extended to 6 h. The relative humidity in the room varied between 63 and 71%. Feed in mash form was offered to the chickens ad libitum, and the feed that was not consumed by the birds was also recorded. Water was offered to the birds using a nipple system. Body weight was registered to determine the performance of the chickens before slaughter and the dissection of organs from the upper part of GIT as a covariate. On day 28 of the experiment, 81 chickens were slaughtered (nine in each treatment) to determine the weight of the individual organs. The proventriculus and gizzard were separated from the gastrointestinal tract crop, and their weight was determined and registered following the removal of digesta.

### 2.2. Chemical Analysis of Feed Nutrients and Structural Components

The content of basic nutrients in the feed components and diets was determined according to the *Official Methods of Analysis* [33]. This included the determination of dry matter (DM, AOAC; 934.01), crude protein (CP, Kjeldahl method, AOAC 984.13), crude ash (CA, AOAC 942.05), ether extract (EE, Soxhlet method, AOAC, 920.39A), crude fibre (CF, Hennenberg and Stohmann method, AOAC 978.10), neutral detergent fibre (NDF, Van Soest method, AOAC 2002:04), acid detergent fibre (ADF, Van Soest method, AOAC 973.18), total dietary fibre (TDF, Asp method, AOAC 991.43), and soluble and insoluble dietary fibre (SDF and IDF, Asp method, AOAC 991.42/43). Hemicellulose was calculated as the difference between NDF and ADF.

### 2.3. Microbial Analysis

During the dissection of the gastrointestinal tract from broiler chickens, the crop and the ileum were collected for microbial analysis (day 28). The collected segments were packed in sterile plastic bags and transported for analysis. In the laboratory, the total aerobic microbial count (TAMC), the total combined yeast and mould count (TYMC), and the counts of *E. coli*, *Lactobacillus* spp., and *Salmonella* sp. were determined and analysed for all diets. Individual microorganisms were inoculated into specific media using selective culture-based plate techniques followed by incubation, after which the counts were determined as CFU · g^−1^.

For analysis, 1 g samples were prepared, to which 9 mL buffered peptone water was added. Serial decimal dilutions were then prepared, which were then inoculated on or in microbiological media. The following groups of microorganisms were determined: TAMC (medium: PN-ISO-4833, 30 °C incubation temperature, 72 h incubation time:), *Lactobacillus* spp. (medium: MRS PN-A-82055-17, 30 °C incubation temperature, 72 h incubation time), *E. coli* (medium: Chromocult coliform agar-Merck, 37 °C incubation temperature, 48 h incubation time), *Salmonella* sp. (medium: VRBG agar PN-A-04023, 37 °C incubation temperature, 24 h incubation time), and TYMC (medium: PN-ISO-7954, 25 °C incubation temperature, 168 h incubation time). After incubation, the microorganism counts were determined and registered.

### 2.4. Statistical Analysis

All obtained numerical data, as mean values for each cage, were evaluated statistically by one-factorial ANCOVA composed of a covariate with a two-ANOVA factorial design using TIBCO Statistica 13.3 software [34]. In the case of organ weights, the covariate was the post-mortem body weight. The covariates in the microbial analysis were the crop pH and the pH in the initial part of the ileum for each gastrointestinal tract region. Microorganism count data were also logarithmically transformed. Differences in the mean values between treatments were evaluated by means of the Duncan test. The statistical significance of differences was determined according to significance levels corresponding to *p* < 0.05 and *p* < 0.01.

## 3. Results

The crude fibre content of diets increases with the amount of oat hull (Table 4). On day 28, the chickens kept on diets with barley grain reached higher (*p* < 0.01) mean body weight (1501 g) than the birds on a diet with 50% wheat (1426 g) and in the control group (1343 g). The addition of oat hull had no influence on the final BW of the birds (*p* > 0.05). The SEM for all observations in the experimental groups was very low and did not exceed 0.02.

The application of wheat or barley in the diet increased (*p* < 0.05) the weight of digesta-free crop in comparison with maize (Table 5). The inclusion of oat hull decreased the crop weight from 6.7 g in the case of birds fed diets without the addition of this component to 5.4% in the treatment of the chickens fed a diet with the addition of 1% OH (*p* < 0.01) and by 5.8% in the case of birds fed a diet with the addition of 3% OH. The proventriculus weight differed significantly between the groups (*p* < 0.01). The highest value was recorded in the case of the barley diet (6.7 g), followed by wheat (6.3 g), and then maize (5.9 g). The use of oat hull caused the proventriculus weight to increase (*p* < 0.01) from 6.0 g in the group fed a diet without the addition of oat hull to 6.5 g when the diet included 1% oat hull. The birds that were kept on a barley diet had a heavier gizzard (28.7 g; *p* < 0.01) than chickens fed with wheat and maize diets (23.8 and 24.8 g, respectively). The addition of oat hull increased (*p* < 0.01) the gizzard weight to 26.3 and 27.3 g (for 1 and 3%, respectively) from 23.7 g for the diet without the addition of OH.

Table 6 presents information about the individual microorganisms in the crop and small intestine. The birds fed a diet containing maize and wheat reached higher (*p* < 0.01) pH values in the crop (5.82 and 5.84, respectively) than the chickens who received a diet containing barley (5.46). The addition of 1 and 3% oat hull also resulted in crop pH values increasing (*p* < 0.01) to 5.74 and 5.75, respectively, from 5.63 for the diet without the addition of structural carbohydrates. The addition of wheat resulted in an increase (*p* < 0.01) in the total aerobic microbial count (6.29 log CFU · g^−1^) compared with barley (5.08 log CFU · g^−1^) and maize (4.62 log CFU · g^−1^). The diet with 1% OH resulted in a significant increase (*p* < 0.01) in the total aerobic bacteria count (5.53 log CFU · g^−1^) compared with both the diet with no addition of OH and the diet with the addition of 3% OH (5.28 and 5.18 CFU · g^−1^, respectively). A higher count of *E. coli* was confirmed in the case of diets containing barley grain (4.21 CFU · g^−1^) and feed with oat hull (4.02–4.03 CFU · g^−1^), but the differences were not statistically significant. The lowest crop *Lactobacillus* spp. count (3.55 log CFU · g^−1^) was noted for the maize diet (*p* < 0.01), and the highest populations of this bacterium were observed in the crop of birds kept on the wheat and barley diets (4.05 and 3.89 CFU · g^−1^, respectively). No *Salmonella* sp. was found in the crop of birds in this experiment.

The analysis of small intestine pH showed no differences in the case of the cereal diet (*p* > 0.05), with an average value of 6.82. The addition of 3% oat hull resulted in a significant increase (*p* < 0.01) to 7.02, while the lowest pH was noted when 1% OH was added to the feed (6.64) with no significant difference compared with the pH of the birds fed diets without OH (6.80). The addition of oat hull resulted in a significant difference in the TAMC count (*p* < 0.05) between the 1% and 3% OH diets (4.01 and 4.54 log CFU · g^−1^, respectively). In the case of cereal grain, a lower *E. coli* count (*p* < 0.01) was noted in the chickens kept on the barley diet (2.69 log CFU · g^−1^) compared with those fed a diet containing wheat and maize (3.41 and 3.45 log CFU · g^−1^, respectively). For the hull factor, a significant difference (*p* < 0.05) was found between the diets containing 1% OH and without the addition of OH (3.32 and 3.04 log CFU · g^−1^, respectively), while the addition of 3% of this component to the diet did not result in significant differences to the other treatments (*p* > 0.05). The addition of 1% OH increased the TYMC (*p* < 0.05) in the intestine compared with the diets lacking OH addition (1.73 and 0.93 log CFU · g^−1^, respectively), while the addition of 3% OH (1.46 log CFU · g^−1^) resulted in no difference with respect to the other treatments (*p* > 0.05). No *Salmonella* sp. was found in the intestine.

The total aerobic bacteria count in the feed was the highest in the feed containing only maize and containing maize together with 1 or 3% OH (1.45–1.40 log CFU · g^−1^, Table 4). The TAMC was lower in the feed for barley than for maize diets, ranging from 1.28 to 1.30 log CFU · g^−1^. The lowest TAMC was recorded in the feeds for wheat diets, ranging from 0.29 to 0.34 log CFU · g^−1^, which was more than four times lower than in the other feed mixtures. The most abundant *Lactobacillus* spp. microflora was also noted in the maize diets (2.11–2.13 log CFU · g^−1^), which was similar to the TAMC determined in the barley diets (1.26–1.28 log CFU · g^−1^) and more than two times lower in wheat diets (0.87–0.89 log CFU · g^−1^). The total yeast and mould count was also highest in the feed of the maize diets (1.86–1.91 log CFU · g^−1^), was a little lower for the wheat feeds (1.76–1.82 log CFU · g^−1^), and was the lowest for the barley diets (1.43–1.46 log CFU · g^−1^). *E. coli* and *Salmonella* sp. were not detected in the experimental feeds.

The correlation coefficients between microorganisms are given in Table 7. Significant negative correlations were confirmed between the TAMC in the diets in comparison to *Lactobacillus* spp. and the crop TAMC (respectively r = −0.907 and r = −0.821), between the TYMC in the diet and *E. coli* in the crop (r = −0.457), and between *Lactobacillus* spp. in feed and crop (r = −0.461). No correlation was found between the feed TAMC and TYMC and feed *Lactobacillus* spp.

Significant positive correlations were confirmed between the TYMC in the feed and *E. coli* in the initial part of the ileum (r = 0.499), between the TYMC in the feed and *Lactobacillus* spp. in the small intestine (r = 0.404), and between the *Lactobacillus* spp. count in the feed and the TYMC in the small intestine (r = 0.409).

A significant (*p* < 0.05) strong negative correlation (r = −0.767) was noted between *Lactobacillus* spp. in the crop and the TAMC in the small intestine. A moderate positive correlation (*p* < 0.05) was observed in the case of the TYMC in the crop and the TAMC in the ileum (r = 0.466). Moderate correlations (*p* < 0.05) were observed between *Lactobacillus* spp. in the crop and *E. coli* in the ileum, and between the TYMC in the crop and *E. coli* in the ileum (negative, r = −0.483 and positive, r = 0.448, respectively). A strong negative correlation (*p* < 0.05) was observed between the *E. coli* count in the crop and *Lactobacillus* spp. in the ileum (r = −0.578), and a strong positive correlation (*p* < 0.05) between the *E. coli* count and the TYMC (r = 0.551).

Very strong positive correlations (*p* < 0.05) were determined between the SDF and hemicellulose content in the diet vs. the TAMC in the crop (r = 0.918 and r = 0.816, respectively; Table 8). Strong correlations (*p* < 0.05) were noted for the amounts of IDF and NDF in the diet and the TAMC in the crop (negative, r = −0.563 and positive, r = 0.663, respectively). A moderate negative (*p* < 0.05) was determined between the TDF in the feed and the TAMC in the crop (r = −0.406). The CF, TDF, IDF, and ADF contents in the feed were moderately positively correlated (*p* < 0.05) with the crop *E. coli* count (0.403 ≤ r ≤ 0.485). The *Lactobacillus* spp. count in the crop also exhibited a moderate positive correlation (*p* < 0.05) with the amounts of SDF, NDF, and hemicellulose in the diet (r = 0.416, 0.425 and 0.442, respectively).

Significant medium negative correlations (*p* < 0.05) were observed for the contents of the CF, TDF, and IDF vs. the *E. coli* count in the ileum (−0.480 ≤ r ≤ −0.452). In the case of the TYMC in the small intestine vs. the SDF and hemicellulose in the diet, a weak negative correlation was registered (r = −0.397 and r = −0.398, respectively).

## 4. Discussion

### 4.1. Components and Chemical Analysis

For the diets in which maize was the main cereal component, it was necessary to exceed a content of 500 g · kg^−1^ in order to maintain the diet formula with maize and soybean meal as standard components only. In the other cases, the basic cereal in the diet (wheat, barley) was added at 500 g · kg^−1^. The addition of 1 and 3% oat hull required the increased use of soybean oil, especially in diets with barley grain, of up to 7% in the case of 3% oat hull, which is the limit during the technological process of production for which the hygiene of the production line can be maintained. Diets were isoenergetic and isoproteic. The necessity of using crude fat, which is a potential source of energy for broiler chickens, increased the level of ether extract determined in the experimental analysis to 80 g · kg^−1^. Berrocoso et al. [35], testing diets with 3% oat hull, corn (206 g · kg^−1^), wheat (314 g · kg^−1^), and soybean oil (50 g · kg^−1^), obtained mixtures with 66.2 g · kg^−1^ ether extract. As predicted, the use of 1 and 3% oat hull increased the share of crude fibre in compound feed mixtures, which was of great importance in diets containing wheat grain when considering the proportion of the SDF fraction in relation to the TDF. Additionally, the presence of 3% oat hull in the diet increased the amount of CF to 37.6, 40.7, and 54.5 g · kg^−1^, respectively, for diets containing maize, wheat, and barley grain as the main grain component. Similar data were obtained by Ben-Mabrouk et al. [36] for brown-egg pullets reared for 15 weeks. The addition of 3% oat hull and 643 g · kg^−1^ corn increased the amount of CF to 40 g · kg^−1^ in diets in weeks 0–6. Barley (358 g · kg^−1^), corn (371 g · kg^−1^), and 3% oat hull offered to birds between week 7 and week 10 increased the amount of CF to 44 g · kg^−1^.

### 4.2. Development of the Upper Part of the Gastrointestinal Tract

The process of upper gastrointestinal tract development is slightly different when using oat hull compared with the application of cereal grains only. The food content does not remain in the crop for a long time, which resulted in lowered weight (*p* < 0.01) compared with the crop of chickens fed with diets without this structural component. Instead, it is more rapidly directed to the gizzard, which significantly increases the weight of both muscle and glandular stomachs. This has also been reported in relation to the increased surface rubbing of food content by the inner kaolin layer in the case of the gizzard, which increases the secretory surface in the case of the proventriculus [37]. Svihus [38] described the rapid increase in the day scale of chickens’ gizzard as a response to the increased amount of insoluble dietary fibre in the diet of broiler chickens (oat hull, sawdust, and sugarcane bagasse), while Kheravii et al. [9] also emphasised that the proventriculus produces more hydrochloric acid as a result of the increased amounts of dietary fibre. Additionally, Sadeghi et al. [26] described that the use of insoluble fibre, in addition to lowering the pH in the upper part of GIT and influencing pathogenic bacteria, also has anticoccidiostatic effects towards *Eimeria* sp. The suboptimal contents of dietary fibre in bird diets cause gizzard erosion and ulceration syndrome [39]. It is also the reason for diminished feed–flow regulation, which affects the excessive filling of the small intestine and reduces nutrient absorption [40].

### 4.3. Microorganisms in the Crop and the Initial Part of the Ileum

The microorganism count analysis was performed using selective culture-based techniques to determine the microbial diversity in the crop and the ileum; however, in this case, it was combined with the covariance analysis of the dietary fibre fraction content. No *Salmonella* sp. bacteria were found in the feed or in the digestive tract up to the initial section of the ileum. Singh and Kim [41] reported that the microbiome, from the crop to the cloaca, inhibits the intestinal content of microorganisms to within 10^11^–10^12^ CFU · g^−1^. Their number increases gradually along with their location in the digestive tract of chickens. Kheravii et al. [9] report that the increased amount of insoluble dietary fibre in the diet increases the intensity of fermentation in the crop, which reduces the pH and increases the activity of pepsin in the small intestine along with the *Clostridium perfringens*, *Escherichia coli*, and *Salmonella* sp. counts. Branton et al. [42] reported the beneficial effect of wood shavings’ insoluble NSP in significantly reducing necrotic enteritis in the intestine.

### 4.4. Total Number of Aerobic Bacteria (TAMC)

In the feed, the main source of the studied bacteria was corn grain, and the counts ranged from 1.40 to 1.45 log CFU · g^−1^, while the lowest TAMC was recorded in mixtures with wheat grain. There were significant differences in the crop TAMC depending on the type of cereal (*p* < 0.01) and the level of OH (*p* < 0.05). A higher TAMC was found in mixtures with a high proportion of wheat grain compared with diets that included barley and corn. The use of oat hull in mixtures increased the TAMC, but only for a 1% addition. Higher OH levels negatively affect the rate of the multiplication of aerobic bacteria in the crop, which in turn confirms that the highest number of these microorganisms in the crop is not influenced by the number of bacteria in the diet, as the correlation coefficient for their number in the feed and the crop is r = −0.907, but rather important in this case is the amount of SDF and hemicellulose in the diet (r = 0.918 and r = 0.816). This phenomenon has a positive effect in the case of changes in the composition of the microflora within the crop, and as confirmed by Singh and Kim [39], the addition of soluble dietary fibre in the diet influences the growth and colonisation of the walls of this organ by promoting the growth of beneficial and probiotic bacteria by providing them with substrates for energy and thus their growth. In the experiment, the TAMC responded most strongly to the proportion of SDF and, to a lesser extent, *Lactobacillus* spp. The use of OH in greater proportions, thus increasing the amount of IDF, resulted in a decrease in the crop TAMC. 

In the initial part of the ileum, the TAMC decreased to an average of 4.31 log CFU · g^−1^, which is supported by the correlation coefficient between the number of bacteria in the crop and ileum, r = −0.186. The addition of a greater amount of OH in feed mixtures (4.54 log CFU · g^−1^) had a significant impact on the TAMC (*p* < 0.05) in this section. Rintilla and Apajalahti [43] described the upper gastrointestinal environment as being more aerobic, which favours the multiplication of aerobic bacteria, which is in line with the values observed in the experiment, where the crop was dominated by aerobic bacteria and relative anaerobes in the form of *Lactobacillus* spp. and several-percent shares of components that are sources of SDF, such as soybean hull, dried beet pulp, and dried citrus pulp [44]. They strengthen the muscle stomach barrier by lowering the pH in order to eliminate pathogenic bacteria already in the upper gastrointestinal tract; this effect is enhanced by the addition of insoluble fibre, which reduces the pH in the upper gastrointestinal tract by the actions of the proventriculus and gizzard as well as the SCFAs that are produced as a result of fermentation in the crop [45,46].

Anaerobic conditions predominate in the lower gastrointestinal tract, and, hence, there is a decrease in the TAMC in the initial part of the ileum as observed in the experiment. As a result, there was also a decrease in the competition for nutrients between the host and the small intestine microflora [9,47]. Hence, the diluent effect of the IDF-rich dietary fibre is of great importance in regulating the microflora in this section of the digestive tract [17].

### 4.5. E. coli

No *E. coli* bacteria were found in the feed mixtures; therefore, the influence of the presence of these bacteria in the feed on the number of all the tested types of bacteria in the crop was not taken into account. The increased amount of oat hull in the diet as a source of IDF was negatively correlated (*p* < 0.05) with the number of *E. coli* in the ileum (r = −0.462), and increased numbers of these bacteria in the goitre reduced the number of *Lactobacillus* spp. in the ileum (*p* < 0.05, r = −0.578). This is in accordance with the observations of Bogusławska-Tryk et al. [46], who, as a result of including lignocellulose in the diet of chickens, noted a reduction in the number of *E. coli* and *Clostridium* sp. in the ileum and ceca.

### 4.6. Lactobacillus *spp.*

The OH had no effect on the number of *Lactobacillus* spp. in the crop (*p* > 0.05). The presence of lactic acid bacteria in the feed was, on average, negatively correlated with their number in the crop (*p* < 0.05 and r = −0.461) and strongly negatively correlated with the TAMC (*p* < 0.05 and r = −0.821). There was no effect of crude and dietary fibre fractions on the number of *Lactobacillus* spp. in the ileum, although Kheravii et al. [9] reported that fibre can stimulate the growth of some species of bacteria, such as *Bifidobacterium* or *Lactobacillus* spp. Similar results were presented by Bogusławska-Tryk et al. [48] in response to an increase in the content of lignocellulose in the diet. Increasing the number of relatively or strictly anaerobic lactic acid bacteria in the crop reduces the total number of aerobic bacteria in the ileum environment, which is also a consequence of decreased oxygen availability (*p* < 0.05 and r = −0.767); it is also negatively correlated with the number of *E. coli* (*p* < 0.05 and r = −0.483). In turn, Mateos et al. [3], apart from stimulating the multiplication of the above groups of bacteria, also added SCFAs, and this action prevented digestive disturbances in broiler chickens. This was confirmed to be due to the phenomenon of competitive exclusion of pathogenic bacteria by lactic acid bacteria described by Nurmi et al. [49].

In addition, the fermentation of dietary fibre by bacteria increases the concentration of SCFAs, which consequently lowers the pH and thus promotes the proliferation of other beneficial groups of bacteria, increasing the integrity of the intestinal immune barrier [50]. Sabour et al. [46] showed an increase (*p* < 0.05) in the intestinal *Lactobacillus* count as a response to the use of insoluble fibre in the diet (12.76 log CFU · g^−1^) compared with soluble fibre (9.98 log CFU · g^−1^).

### 4.7. Total Yeast and Moulds Content (TYMC)

The application of 3% OH decreased the TYMC in the crop of chickens at 28 days old to 1.92 log CFU · g^−1^ compared with the mixtures prepared without the use of this structural component. The TYMC in the diet did not affect their content in the crop of chickens, but it decreased the number of *E. coli* in the crop. The TYMC in the feed was also positively correlated (*p* < 0.05) with the number of *E. coli* and *Lactobacillus* spp. in the ileum (r = 0.499 and r = 0.404, respectively). Additionally, an increase in the soluble fractions of dietary fibre in the feed SDF and hemicellulose negatively affected the TYMC in the ileum. This indicates that mould species may degrade IDF to some extent or use substrates resulting from the degradation of this fibre by *Lactobacillus* spp. However, the correlation coefficients of IDF and NDF in the feed and the TYMC are low, which is in agreement with the observations of Singh and Kim [41], and it is assumed that bacteria of the genus *Lactobacillus* spp. are responsible for the breakdown of fibre in diets where they are found at higher contents. At the same time, the benefits of using fibre are revealed only when included at up to a 3% proportion of the total diet [51].

## 5. Conclusions

A diet supplemented with 3% oat hull decreased the TYMC in the crop (ca. 17%), while the addition of 1% OH increased the TYMC by ca. 86% in the ileum in relation to diets with no OH, suggesting that moulds and yeast could help digest dietary fibre in the ceca to a greater degree than in the crop. Feeds containing wheat grain increased the TAMC (ca. 36%) and *Lactobacillus* spp. (ca. 14%) counts in the crop compared with diets including maize, with the main reason for the growth of microorganisms being the ratio of the amount of the SDF and hemicellulose present in a diet to the TAMC (r = 0.918 and 0.816, respectively) and *Lactobacillus* spp. (r = 0.416 and 0.442, respectively). On the other hand, the presence of IDF in the diets decreased the TAMC (r = −0.563) and increased the *E. coli* (r = 0.403) counts in the upper part of GIT. Barley diets decreased the level of *E. coli* in the ileum (ca. 21%) compared with maize and wheat diets, mainly as a result of the increased amount of IDF (r = −0.462). In the ileum, a decreased count of TYMC was also observed as an effect of the SDF and hemicelluloses present in diets (respectively r = −0.397 and −0.398). The results indicate that the addition of insoluble fibre to chicken diets may be reasonable for maintaining microbial balance in the crop and ileum. However, sources of small amounts of SDF can be taken into consideration (sugar beet pulp, soybean hull, and citrus pulp up to 3%) for increasing the TAMC and *Lactobacillus* spp. bacteria in the crop in order to prevent colonisation by pathogenic bacteria already in the upper GIT.

## Figures and Tables

**Table 1 animals-12-02721-t001:** Experimental design.

First Factor: Cereal Grain (*n* = 27)								
Control			Wheat			Barley		
9 replications			9 replications			9 replications		
**Second factor: Oat hull (*n* = 27)**								
C0 *	C1	C3	W0 *	W1	W3	B0 *	B1	B3
3 repli.	3 repli.	3 repli.	3 repli.	3 repli.	3 repli.	3 repli.	3 repli.	3 repli.

* C0, C1, C3–control diet with 0–3% oat hull; W0, W1, W3–wheat diet with 0–3% oat hull; B0, B1, B3 barley diet with 0–3% oat hull.

**Table 2 animals-12-02721-t002:** Composition of structural carbohydrates in components (g · kg^−1^).

Specification	Maize	Wheat	Barley	Soybean Meal	Oat Hull
Crude fibre	28.1	37.3	67.8	34.9	278.1
TDF	150.9	141.1	253.7	328.4	661.9
SDF	1.53	30.9	11.7	30	9.61
IDF	149.5	110.1	240.7	298.4	652.3
NDF	135.3	231.8	224.6	97.1	639.1
ADF	37.3	45.2	66.5	47.9	342.4
Hemicellulose	97.8	186.6	158.1	49.1	296.7

**Table 3 animals-12-02721-t003:** Ingredient composition of diets (g · kg^−1^).

Ingredients	Maize OH * 0%	Maize OH 1%	Maize OH 3%	Wheat OH 0%	Wheat OH 1%	Wheat OH 3%	Barley OH 0%	Barley OH 1%	Barley OH 3%
Maize	559	547	520	81	68	41	7	-	-
Wheat	-	-	-	500	500	500	-	-	-
Barley	-	-	-	-	-	-	500	500	500
Oat hull	-	10	30	-	10	30	-	10	30
Soybean oil	28	30	33	41	43	47	69	70	70
Soybean meal	369	370	374	334	336	339	382	378	358
Dicalcium phosphate	21.11	20.03	20.37	19.26	18.50	18.49	17.26	17.15	17.18
Limestone	2.36	2.38	2.11	3.11	3.08	3.02	3.95	3.99	4.04
NaCl	3.42	3.47	3.37	3.67	3.52	3.54	3.62	3.69	3.61
DL-Methionine 98%	2.11	2.12	2.15	2.29	2.19	2.21	2.17	2.17	2.17
L-Lysine HCL 78%	-	-	-	0.67	0.71	0.74	-	-	-
Cr_2_O_3_	5	5	5	5	5	5	5	5	5
Premix DKA-s **	10	10	10	10	10	10	10	10	10

* OH–oat hull; ** Premix provided the following per kilogram of diet: vitamin A 10,000 IU, vitamin D_3_ 2000 IU, vitamin E 20 mg, vitamin K 3 mg, vitamin B_1_ 2.5 mg, vitamin B_6_ 0.4 mg, vitamin B_12_ 0.015 mg, nicotinic acid 60 mg, pantothenic acid 8 mg, folic acid 1.2 mg, choline chloride 450 mg, DL-Methionine 1.0 mg, Mn 74 mg, Fe 30 mg, Zn 45 mg, Cu 4 mg, Co 0.4 mg, I 0.3 mg.

**Table 4 animals-12-02721-t004:** Chemical, structural, and microbial composition of experimental diets.

Specification	Maize OH 0%	Maize OH 1%	Maize OH 3%	Wheat OH 0%	Wheat OH 1%	Wheat OH 3%	Barley OH 0%	Barley OH 1%	Barley OH 3%
Metabolizable energy (MJ·kg^−1^)	12.24	12.25	12.21	12.18	12.18	12.16	12.24	12.23	12.22
**Nutrients (g · kg^−1^)**									
Dry matter	926.9	922.4	918.8	915.6	899.9	900.8	909.9	919.8	901.0
Crude protein	225.3	224.6	218.0	211.7	211.2	212.8	225.6	230.7	217.8
Ether extract	49.3	52.8	54.9	53.2	57.1	58.4	83.8	84.2	78.0
Crude ash	41	41	41	39	39	40	51	51	51
**Structural components (g · kg^−1^)**									
Crude fibre	29.4	31.2	36.7	32.0	35.1	40.7	48.1	50.4	54.5
TDF *	203.2	212.2	220.0	191.6	199.1	209.3	248.4	253.0	263.5
SDF *	12.1	12.1	12.2	25.8	26.1	26.3	17.0	17.4	16.9
IDF *	191.1	200.1	207.8	165.8	173.0	183.0	231.4	235.6	246.6
NDF *	109.9	114.5	125.7	158.9	161.0	175.8	151.1	156.4	160.8
ADF *	38.4	42.4	47.1	41.2	44.3	50.1	51.5	55.5	61.5
Hemicelluloses	71.5	72.1	78.6	117.7	116.7	125.7	99.6	100.9	106.5
**Count of microorganisms in feed (log CFU · g^−1^)**									
TAMC *	1.45	1.44	1.40	0.34	0.33	0.29	1.29	1.28	1.30
*Lactobacillus* spp.	2.13	2.12	2.11	0.89	0.89	0.87	1.26	1.26	1.28
TYMC *	1.91	1.90	1.86	1.82	1.80	1.76	1.46	1.44	1.43
*E.coli*	-	-	-	-	-	-	-	-	-
*Salmonella* sp.	-	-	-	-	-	-	-	-	-

* TDF-total dietary fibre, SDF-soluble dietary fibre, IDF-insoluble dietary fibre, NDF-neutral dietary fibre, ADF–acid dietary fibre, TAMC-total aerobic microbial count, TYMC-total yeast and mould count, OH-oat hull.

**Table 5 animals-12-02721-t005:** Final body weight and weight of organs from the upper part of GIT (*n* = 27, in g).

Specification	Final Body Weight *	Weight of		
		Crop without Digesta	Proventriculus	Gizzard
**Cereal**				
Maize	**1343 ^C^**	5.5 ^b^	5.9 ^C^	24.8 ^B^
Wheat	**1426 ^B^**	6.1 ^a^	6.3 ^B^	23.8 ^B^
Barley	**1501 ^A^**	6.2 ^a^	6.7 ^A^	28.7 ^A^
**Hull**				
0%	**1422**	6.7 ^Aa^	6.0 ^B^	23.7 ^B^
1%	**1437**	5.4 ^Bb^	6.5 ^Aa^	26.3 ^A^
3%	**1404**	5.8 ^b^	6.3 ^Ab^	27.3 ^A^
SEM	**0.018**	0.209	0.090	0.555
** *p* ** **-value**				
Cereal	**0.000**	0.043	0.045	0.000
Hull	**0.114**	0.019	0.001	0.000
**Covariate ***	-	**0.086**	**0.011**	**0.269**

Means in the same row with different superscripts a, b are significantly different with *p* ≤ 0.05 and A, B, C with *p* ≤ 0.01. Covariation significant with *p* < 0.05. * Covariate.

**Table 6 animals-12-02721-t006:** Counts of microorganisms in crop and small intestine (log CFU · g^−1^).

Specification	Crop						Ileum					
	pH *	TAMC	*E. coli*	TYMC	*Lactobacillus* spp.	*Salmonella* sp.	pH *	TAMC	*E. coli*	TYMC	*Lactobacillus* spp.	*Salmonella* sp.
**Cereal**												
Maize	**5.82 ^A^**	4.62 ^C^	3.81	2.19	3.55 ^Bb^	-	**6.83**	4.49	3.41 ^A^	1.75	2.30	-
Wheat	**5.84 ^A^**	6.29 ^A^	3.85	2.09	4.05 ^A^	-	**6.79**	4.15	3.45 ^A^	1.03	1.84	-
Barley	**5.46 ^B^**	5.08 ^B^	4.21	2.06	3.89 ^Aa^	-	**6.84**	4.29	2.69 ^B^	1.34	1.44	-
**Oat hull**												
0%	**5.63 ^B^**	5.28 ^a^	3.82	2.30 ^A^	3.80	-	**6.80 ^AB^**	4.38 ^ab^	3.04 ^b^	0.93 ^a^	1.61	-
1%	**5.75 ^A^**	5.53 ^a^	4.03	2.12 ^ABa^	3.86	-	**6.64 ^B^**	4.01 ^a^	3.32 ^a^	1.73 ^b^	2.12	-
3%	**5.74 ^A^**	5.18 ^b^	4.02	1.92 ^Bb^	3.84	-	**7.02 ^A^**	4.54 ^b^	3.18 ^ab^	1.46 ^ab^	1.84	-
SEM	**0.042**	0.151	0.082	0.049	0.088	-	**0.049**	0.149	0.129	0.138	0.161	-
** *p* ** **-value**												
Cereal	**0.000**	0.000	0.477	0.115	0.001	-	**0.424**	0.423	0.002	0.070	0.097	-
Hull	**0.002**	0.012	0.063	0.020	0.106	-	**0.000**	0.020	0.048	0.034	0.339	-
**Covariate ***	-	**0.021**	**0.015**	**0.120**	**0.000**	-	-	**0.006**	**0.000**	**0.619**	**0.479**	-

Means in the same row with different superscripts a, b are significantly different with *p* ≤ 0.05 and A, B with *p* ≤ 0.01. * Covariation significant with *p* < 0.05. TAMC-total aerobic microbial count, TYMC-total yeast and mould count.

**Table 7 animals-12-02721-t007:** Correlation between count of microorganisms (*n* = 27).

Feed	Crop			
	TAMC	*E. coli*	*Lactobacillus* spp.	TYMC
TAMC	−0.907 *	0.103	−0.375	0.089
*Lactobacillus* spp.	−0.821 *	−0.144	−0.461 *	0.188
TYMC	0.038	−0.457 *	−0.188	0.229
**Feed**	**Ileum**			
	**TAMC**	** *E. coli* **	** *Lactobacillus* ** **spp.**	**TYMC**
TAMC	0.154	−0.241	0.073	0.354
*Lactobacillus* spp.	0.181	0.084	0.306	0.409 *
TYMC	0.030	0.499 *	0.404 *	0.086
**Crop**	**Ileum**			
	**TAMC**	** *E. coli* **	** *Lactobacillus* ** **spp.**	**TYMC**
TAMC	−0.186	0.184	−0.188	−0.222
*Lactobacillus* spp.	−0.767 *	−0.483 *	−0.177	−0.026
*E.coli*	0.129	−0.127	−0.578 *	0.551 *
TYMC	0.466 *	0.448 *	−0.298	0.126

* Significant correlation *p*-value < 0.05.

**Table 8 animals-12-02721-t008:** Correlation between dietary/crude fibre in feed and count of microorganisms in crop of broiler chickens (*n* = 27).

Specification	Crop			
	TAMC	*E. coli*	TYMC	*Lactobacillus* spp.
Crude Fibre	−0.083	0.485 *	−0.363	0.163
TDF	−0.406 *	0.443 *	−0.278	0.018
SDF	0.918 *	−0.008	−0.129	0.416 *
IDF	−0.563 *	0.403 *	−0.225	−0.073
NDF	0.663 *	0.234	−0.376	0.425 *
ADF	−0.102	0.420 *	−0.496 *	0.156
Hemicelluloses	0.816 *	0.122	−0.261	0.442 *
**Specification**	**Ileum**			
	**TAMC**	** *E. coli* **	**TYMC**	** *Lactobacillus* ** **spp.**
Crude Fibre	0.028	−0.452 *	0.000	−0.369
TDF	0.056	−0.480 *	0.142	−0.293
SDF	−0.170	0.125	−0.397 *	−0.164
IDF	0.087	−0.462 *	0.213	−0.231
NDF	−0.101	−0.136	−0.330	−0.314
ADF	0.028	−0.382	0.030	−0.271
Hemicelluloses	−0.129	−0.021	−0.398 *	−0.271

* Significant correlation *p*-value < 0.05.

## Data Availability

The data presented in this study are available on request from the corresponding author.

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
