# Peer review of "Effect of Oat Hull as a Source of Insoluble Dietary Fibre on Changes in the Microbial Status of Gastrointestinal Tract in Broiler Chickens"

_animals, 2022, doi:10.3390/ani12192721_

Round 1

Author Response

Anna Szuba-Trznadel

Reviewer 2 Report

The paper investigated an interesting topic very important for poultry nutrition and production, and this research fits well within the overall scope of the section Animal Nutrition of Animals.

However, before a possible acceptance of the manuscript, a huge work of revision is need. 

In the present form, the manuscript is quite hard to read and follow. Thus, I suggest to revise the full-text in the first istance for English language. It is unuseful at this stage to report each single language correction.

Moreover, the Abstract is quite confusing: it should be reduced and it needs a conclusive sentence stressing breafly the overall findings of the study and their usefulness and application (this is the same for Conclusion section).

Also, it is necessary to check if ALL references have been cited in the references list or reported in the full-text...moreover, some additional references of recently published paper may add value to the Introduction and Discussion sections.

Finally, the Tables result quite harrd to follow.

So, I recommend major revision before a possible acceptance of this paper.

Author Response

Anna Szuba-Trznadel

Round 2

Reviewer 1 Report

The article was improved and no serious flaws remains.

Reviewer 2 Report

The Authors have done a good work of revision.